# Attitudes of Medical Students toward COVID-19 Vaccination: Who Is Willing to Receive a Third Dose of the Vaccine?

**DOI:** 10.3390/vaccines9111295

**Published:** 2021-11-08

**Authors:** Norio Sugawara, Norio Yasui-Furukori, Atsuhito Fukushima, Kazutaka Shimoda

**Affiliations:** 1Health Services Center for Students and Staff, Dokkyo Medical University, 880 Kitakobayashi, Mibu 321-0293, Tochigi, Japan; 2Department of Psychiatry, Dokkyo Medical University School of Medicine, 880 Kitakobayashi, Mibu 321-0293, Tochigi, Japan; furukori@dokkyomed.ac.jp (N.Y.-F.); shimoda@dokkyomed.ac.jp (K.S.); 3Department of Infection Control and Clinical Laboratory Medicine, Dokkyo Medical University School of Medicine, 880 Kitakobayashi, Mibu 321-0293, Tochigi, Japan; atsufuku@dokkyomed.ac.jp

**Keywords:** COVID-19, SARS-CoV-2, vaccine, hesitancy, medical student, attitudes

## Abstract

Medical students may come in contact with individuals infected with COVID-19 in their clinical rotations. A high level of acceptance of vaccination is needed for them to protect their health and the health of patients from this disease. The objectives of this study were to (1) obtain information on medical students’ attitudes toward COVID-19 vaccination, (2) assess factors associated with students’ attitudes, and (3) identify predictors of their willingness to receive a third dose of the COVID-19 vaccine. Using a cross-sectional design, we conducted a questionnaire survey of medical students in July 2021. For this survey, we employed a 15-item questionnaire specifically developed to assess the students’ attitudes toward COVID-19 vaccination. Of the 742 distributed questionnaires, 496 (294 males and 202 females) were completed. Among all the participants, 89.1% (442/496) received the second dose of the vaccine, and 90.7% (450/496) indicated that they would hypothetically receive the COVID-19 vaccine in the future. Furthermore, 84.5% (419/496) of all the participants were willing to receive a third dose of the vaccine. Regarding willingness to receive a third dose of the COVID-19 vaccine, multiple logistic regression models showed that students’ grade and their responses to Q1 (positive attitude toward vaccination), Q9 (belief in the protection offered by COVID-19 vaccination), Q10 (concern about the excessively rapid development of COVID-19 vaccines), Q12 (need for aspects of pre-pandemic life), and Q14 (concern about the sustainability of immunity) had significant associations with this outcome. Confidence in vaccines, relaxation of mobility restrictions, and concern about the sustainability of immunity motivate willingness to receive a third dose of the COVID-19 vaccine in medical students.

## 1. Introduction

Since coronavirus disease 2019 (COVID-19), caused by severe acute respiratory syndrome coronavirus 2 (SARS-CoV-2), was first described based on a cluster of cases in China, there have been more than 236 million confirmed cases of COVID-19, including 4.8 million deaths, and the disease has caused an ongoing global pandemic [1]. Preventive measures such as social distancing, quarantining, and wearing masks, have become an essential part of daily life, and this pandemic has affected a wide range of people’s lives, including mental, physical, and social aspects [2,3,4,5]. To eliminate this pandemic, widespread vaccination against COVID-19 has been regarded as a promising measure.

In Japan, COVID-19 vaccination was launched on 17 February 2021 for healthcare workers; the first vaccine approved in Japan, Comirnaty (produced by Pfizer/BioNTech), was based on the use of nucleoside-modified RNA (modRNA), which encodes the spike protein found on the surface of SARS-CoV-2 [6,7]. Medical students in Japan are not involved in clinical practice against COVID-19. However, in clinical rotations, students may come in contact individuals infected with this disease, such as healthcare workers [8]. A high level of acceptance of vaccination is needed for medical students to protect their health and the health of patients from COVID-19 [9,10]. Vaccination is successful only when there are high rates of acceptance and coverage. Healthcare workers are considered an important source of information on vaccination for the general population [11,12], and medical students with a positive attitude toward vaccines might encourage widespread vaccination [9].

Recently, the need for a third dose of the COVID-19 vaccine has been discussed due to the threat of the B.1.617.2 (delta) variant and the need for the protection of immunocompromised individuals [13]. A three-dose regimen could reduce the risk of COVID-19 infection and reverse the waning of neutralizing antibodies after two doses of the vaccine [14,15,16]. However, the best timing of a third dose or boosting with heterologous vaccines is controversial [13,17,18]. Furthermore, there are still many countries facing very large waves of the disease with large unvaccinated populations. In addition, rich countries’ three-dose booster regimens could impede global vaccination [19]. At present, the administration of a third COVID-19 vaccine dose is still controversial.

The objectives of this investigation were to (1) obtain information on medical students’ attitudes toward COVID-19 vaccination, (2) assess factors associated with students’ attitudes, and (3) identify predictors of their willingness to receive a third dose of the COVID-19 vaccine. To the best of our knowledge, this article presents the first study on the willingness to receive a third dose of the COVID-19 vaccine.

## 2. Methods

### 2.1. Participants and Measures

Based on the Health Services Center for Students and Staff (HSCSS) database, more than 80% (648/742) of medical students at Dokkyo Medical University completed a two-dose regimen of the Comirnaty vaccine between April and May in 2021. To assess the attitudes of medical students toward COVID-19 vaccination, the HSCSS conducted a questionnaire survey for medical students at the university in July 2021. For this survey, a brief 15-item questionnaire based on the relevant literature was developed. The surveys were distributed to all medical students at Dokkyo Medical University School of Medicine. Of the 742 distributed questionnaires, 496 (294 males and 202 females) were completed. The response rates among students from Grade 1 to Grade 6, were 73.8% (93/126), 73.2% (90/123), 68.6% (83/121), 69.5% (89/128), 61.2% (63/103), and 55.3% (78/141), respectively. The detailed questions are summarized in Table 1. Students rated the item regarding their attitudes toward COVID-19 vaccination on a 4-point scale (1 = strongly agree, 2 = agree, 3 = disagree, 4 = strongly disagree). Those answering 1 or 2 were grouped as answering “agree”, while those who answered 3 or 4 were considered as answering “disagree”. Sociodemographic data (age, grade, and sex) and history of allergic reaction, anaphylaxis, asthma, atopic dermatitis, and COVID-19 vaccination were obtained from the HSCSS database. Furthermore, we classified students by status of COVID-19 vaccination and willingness to receive the vaccine (Q15) (Figure 1). Students who were unvaccinated or dropped out after the first dose of the COVID-19 vaccine and were willing to receive the vaccine in the future were classified as Group A. On the other hand, those who completed the second dose of the COVID-19 vaccine and were not willing to receive the vaccine in the future were classified as Group B. In addition, those who completed the second dose of the COVID-19 vaccine and were willing to receive the vaccine as the third dose in the future were classified as Group C.

### 2.2. Ethics

This protocol received approval from the Ethics Committee of Dokkyo Medical University School of Medicine (Approval number: 2021–016), and it conformed to the provisions of the Declaration of Helsinki. The requirement for written informed consent was waived by the Ethics Committee since the study involved record review only. However, information on the study was disclosed on our web page, and students were free to opt out.

### 2.3. Statistical Analysis

Descriptive analyses of the demographic and clinical characteristics were performed. To compare the main variables between groups, unpaired Student’s t-tests were performed to analyze the continuous variables, and chi-square tests were performed to analyze the categorical variables. The data are presented as the means ± SDs or percentages. Multivariate logistic regression analysis with a forward selection method was performed to assess the effects of the students’ attitudes (Q1–Q14 for students) and covariates (age, grade, sex, and history of allergic reaction, anaphylaxis, asthma, and atopic dermatitis) on the completion of COVID-19 vaccination with a two-dose regimen. In another analysis, we employed multivariate logistic regression analysis with a forward selection method to assess the effects of the students’ attitudes and the same covariates on the willingness to receive the COVID-19 vaccine in the future (Q15).

Furthermore, we also conducted multivariate logistic regression analysis with a forward selection method to assess the effects of students’ attitudes and the same covariates on the changing willingness to receive the COVID-19 vaccine in the future (Groups A and B). Regarding those who were willing to receive a third dose of the COVID-19 vaccine (Group C), multivariate logistic regression analysis with a forward selection method was employed as the dependent variable and students’ attitudes and the same covariates were employed as independent variables. A value of *p* < 0.05 was considered significant. The data were analyzed using SPSS software for Windows (Version 27).

## 3. Results

Among all the participants, 90.7% (450/496) indicated that they would hypothetically receive the COVID-19 vaccine in the future (Q15) (Table 1). The mean age and grade of all participants were 21.1 ± 2.5 years and 3.4 ± 1.7, respectively. In the analysis of the influence of students’ characteristics (Table 2) on their willingness to receive the COVID-19 vaccine in the future (Q15), no differences in any other characteristics were observed.

To assess the influence of attitudes (Q1–Q14) toward COVID-19 vaccination as independent variables of students’ completion of COVID-19 vaccination with a two-dose regimen, we performed a multivariate logistic regression analysis with a forward selection method. Higher grade and students’ responses to Q1 (positive attitude toward vaccination), Q4 (effect of parents’ opinions), Q10 (concern about the excessively rapid development of COVID-19 vaccines), and Q12 (need for aspects of pre-pandemic life) showed significant associations. In an analysis of students’ willingness to receive the COVID-19 vaccine in the future as a dependent variable, patients’ responses to Q1, Q9 (belief in the protection offered by COVID-19 vaccination), Q10, Q12, and Q14 (concern about the sustainability of immunity) were significantly associated with willingness (Table 3).

Table 4 shows the factors associated with the changed willingness to receive the COVID-19 vaccine. Regarding Group A, responses to Q10 and Q14 showed a significant association. In addition, students’ grade and responses to Q2 (fear of needles/shots), Q4, and Q12 were significantly associated in Group B.

Regarding willingness to receive a third dose of the COVID-19 vaccine, multiple logistic regression models showed that students’ grade and responses to Q1, Q9, Q10, Q12, and Q14 had significant associations with this outcome (Table 5).

## 4. Discussion

This study was conducted to investigate the attitudes of medical students toward COVID-19 vaccination. In our survey, almost all the respondents (98.4%; 488/496) stated that they approved of vaccinations in principle (Q1), and most of them (92.8%; 460/496) considered COVID-19 vaccination to be necessary for them to travel or go out to eat and drink as in pre-pandemic life (Q12). Among the respondents, 89.1% (442/496) had received the second dose of the vaccine, and 90.7% (450/496) indicated that they would hypothetically receive the COVID-19 vaccine in the future. Furthermore, 84.5% (419/496) of all the participants were willing to receive a third dose of the vaccine. Although the majority of the students (75.6%; 375/496) stated that the vaccines provide a high degree of protection against COVID-19 (Q9), 67.3% (334/496) were concerned about the sustainability of immunity by the vaccine (Q14). In addition, a nonnegligible number of the students (46.0%; 228/496) stated that the vaccine was developed too rapidly (Q10).

Vaccine hesitancy, which is defined as a delay in acceptance or refusal of vaccinations even though vaccination services are made available, poses a barrier to achieving widespread vaccination against COVID-19. A previous international meta-analysis of the proportion of vaccine hesitancy among the general population suggested that 20% of the participants intended to refuse COVID-19 vaccination [20]. Regarding medical students, the rate of vaccine hesitancy reported in previous studies has varied considerably due to the different definitions of the concept or timing of investigation in each study [9,10,21,22,23]. The availability of vaccination, or the percentage of vaccinated students might be associated with vaccination hesitancy. Several studies have shown that medical students express higher intentions to be vaccinated than the general population [24], nonmedical students [25,26], nursing students [27], and dental students [28]. A recent study conducted in Japan showed that 14 to 16% of the younger (15-39 years old) subgroup expressed COVID-19 vaccine hesitancy in terms of their hypothetical intention to be vaccinated and that vaccination intention was associated with several factors, including younger age, female sex, and living alone [29]. In our results, 10.9% (54/496) of the students did not complete the second dose of the vaccine. Although hypothetical intention is not a real-life decision, the rate of vaccine hesitancy in the general population seems to be higher than that in medical students who were not vaccinated twice.

Fear of vaccination side effects has been reported to be associated with COVID-19 vaccine hesitancy in medical students as well as the general population [9,26,29,30]. In our analysis, concern about the side effects of COVID-19 vaccination (Q8) was not associated with students’ attitudes toward or completion of vaccination. A possible explanation is that the majority of our participants received the second dose of the vaccine, and their experience might have decreased their concern about the side effects of COVID-19 vaccination. However, concern about the rapid development of COVID-19 vaccines (Q10) still had a significant impact on students’ decision making regarding COVID-19 vaccinations.

Few studies have focused on the effect of parents’ opinions on medical students’ decision making regarding COVID-19 vaccinations. We found significant associations between the effect of parents’ opinion (Q4) and COVID-19 vaccine hesitancy in medical students. Although recommendation from family or friends was indicated as a reason for both receiving and not receiving vaccines [29], it is difficult to explain the association between the effect of parents’ opinion and COVID-19 vaccine hesitancy in medical students. The association found in Q4 might be related to a cultural issue.

Fear of needles has been associated with COVID-19 vaccine hesitancy [23,31] in nonmedical student populations, and our results concerning Q2 might support previous findings. However, our findings also indicated that even students with a fear of needles received a second dose of the COVID-19 vaccine. In our results, students in higher grades had lower vaccine hesitancy. The amount of medical education and need for protection against COVID-19 in clinical rotations might motivate their willingness [32].

Most of our participants considered COVID-19 vaccination to be necessary to resume aspects of pre-pandemic life (Q12). Mobility restrictions were enforced by the government to abate the spread of COVID-19, resulting in motivation to resume going-out activities after the end of the pandemic [33]. However, even for vaccinated individuals, basic preventive measures such as wearing masks are needed to prevent COVID-19 vaccine breakthrough infections for a while [13].

Several limitations of this study should be acknowledged. First, our study is limited by the fact that students were asked about behavioral intention to receive COVID-19 vaccination in the future. The responses of participants may not reflect implementation intention when faced with the decision of actual vaccination. Second, the questionnaires were not standardized, which could lead to problems in interpreting the results. Furthermore, we employed a Likert scale, which leaves room for a subjective overevaluation of the self-perceived degree of comprehension and knowledge. Third, several potential confounding factors, such as depressive symptoms and psychological distress, which may contribute to willingness to receive the COVID-19 vaccine, were not assessed in this study. Fourth, the study population consisted of students from one medical university; therefore, it may not reflect the characteristics of all Japanese medical students.

## 5. Conclusions

Most of the medical students completed the second dose of the COVID-19 vaccine and were willing to receive a third dose. Confidence in vaccines, relaxation of mobility restrictions, and concern about the sustainability of immunity motivated willingness to receive the third dose of the COVID-19 vaccine in medical students. Our findings provide key information related to medical student vaccination in Japan and education about COVID-19 vaccination.

## Figures and Tables

**Figure 1 vaccines-09-01295-f001:**
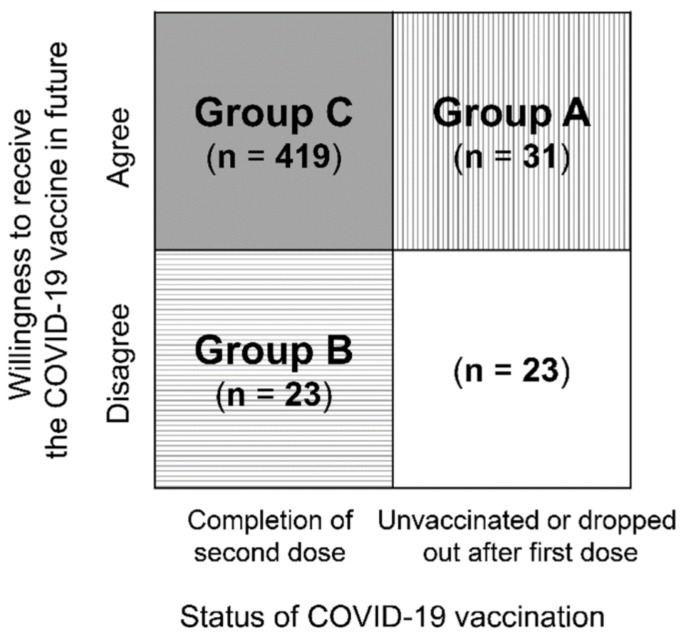
Status of COVID-19 vaccination and willingness to receive the vaccine. **Note**. The box shaded with vertical lines represents students (Group A) who were unvaccinated or dropped out after first dose of the COVID-19 vaccine but who were willing to receive the vaccine in future. On the other hand, the box shaded with horizontal lines represents those (Group B) who completed the second dose of the COVID-19 vaccine but were not willing to receive the vaccine in future. Furthermore, the box shaded with a gray background represents those (Group C) who were willing to receive a third dose of the COVID-19 vaccine.

**Table 1 vaccines-09-01295-t001:** Attitudes of medical students toward COVID-19 vaccination (n = 496).

		Strongly						Srtongly	
		Agree	(n)	Agree	(n)	Disagree	(n)	Disagree	(n)
Q1	In principle, I approve of vaccinations.	51.2%	(254)	47.2%	(234)	1.2%	(6)	0.4%	(2)
Q2	I have a fear of needles/shots.	5.0%	(25)	24.8%	(123)	40.3%	(200)	29.8%	(148)
Q3	If I get COVID-19, I will develop severe symptoms.	3.0%	(15)	27.6%	(137)	61.9%	(307)	7.5%	(37)
Q4	Parents’ opinion affected my decision of whether to receive the COVID-19 vaccine.	13.1%	(65)	32.7%	(162)	29.8%	(148)	24.4%	(121)
Q5	Friends’ opinion affected my decision of whether to receive the COVID-19 vaccine.	6.3%	(31)	33.3%	(165)	33.1%	(164)	27.4%	(136)
Q6	News media affected my decision of whether to receive the COVID-19 vaccine.	4.0%	(20)	33.1%	(164)	35.3%	(175)	27.6%	(137)
Q7	Being free of charge affected my decision of whether to receive the COVID-19 vaccine.	25.2%	(125)	34.1%	(169)	23.4%	(116)	17.3%	(86)
Q8	In general, the COVID-19 vaccine has apparent side effects.	22.8%	(113)	51.4%	(255)	22.8%	(113)	3.0%	(15)
Q9	The vaccines provide a high degree of protection against COVID-19.	11.9%	(59)	63.7%	(316)	22.0%	(109)	2.4%	(12)
Q10	The COVID-19 vaccine was developed too rapidly.	10.3%	(51)	35.7%	(177)	48.2%	(239)	5.8%	(29)
Q11	I have received sufficient information about COVID-19 vaccination.	7.3%	(36)	49.6%	(246)	38.9%	(193)	4.2%	(21)
Q12	COVID-19 vaccination is needed for me to travel or go out to eat and drink as in pre-pandemic life.	44.0%	(218)	48.8%	(242)	6.0%	(30)	1.2%	(6)
Q13	The COVID-19 pandemic will continue for a long time.	26.2%	(25)	62.7%	(25)	10.1%	(25)	1.0%	(25)
Q14	I have concerns about the sustainability of immunity by the COVID-19 vaccine.	9.9%	(49)	57.5%	(285)	29.0%	(144)	3.6%	(18)
Q15	I am willing to get the COVID-19 vaccine in the future.	40.7%	(202)	50.0%	(248)	7.7%	(38)	1.6%	(8)

**Abbreviations**: COVID-19, Coronavirus disease 2019.

**Table 2 vaccines-09-01295-t002:** Characteristics according to willingness to receive the COVID-19 vaccine in future.

	Students’ Willingness to Receive					
	Yes			No		
Age	21.1 ± 2.4			20.8 ± 2.6		
Grade	3.4 ± 1.7			3.0 ± 1.6		
Sex (male)	58.9%	(264/450)		65.2%	(30/46)	
Past history of allergic reaction due to						
food	11.5%	(50/435)		6.7%	(3/45)	
medication	5.1%	(22/435)		2.2%	(1/45)	
animal	5.5%	(24/436)		4.4%	(2/46)	
pollen from plants and trees	25.1%	(109/437)		20.0%	(9/47)	
house dust mites	9.9%	(43/438)		6.7%	(3/45)	
unknown allergen	3.9%	(17/435)		4.4%	(2/46)	
Past history of anaphylaxis	0.5%	(2/435)		0.0%	(0/435)	
Past history of asthma	3.4%	(15/435)		2.2%	(1/45)	
Past history of atopic dermatitis	2.8%	(12/435)		0.0%	(0/435)	

**Table 3 vaccines-09-01295-t003:** Factors associated with completion of COVID-19 vaccination with a two-dose regimen and willingness to receive the COVID-19 vaccine in the future.

	Odds Ratio	95%		CI	Wald Value	*p* Value
Completion of COVID-19 vaccination with a two-dose regimen						
Grade	1.57	1.24	-	1.98	13.71	<0.001
Q1	28.66	2.71	-	303.45	7.77	0.005
Q4	0.41	0.21	-	0.83	6.24	0.013
Q10	0.47	0.24	-	0.91	5.05	0.025
Q12	10.49	4.09	-	26.89	23.94	<0.001
Willingness to receive the COVID-19 vaccine in future						
Q1	8.61	1.44	-	51.55	5.56	0.018
Q9	2.52	1.23	-	5.18	6.32	0.012
Q10	0.26	0.12	-	0.55	12.25	<0.001
Q12	5.14	2.18	-	12.09	14.02	<0.001
Q14	2.53	1.22	-	5.28	6.15	0.013

Multivariate logistic regression analysis using a forward selection method was carried out with completion of COVID-19 vaccination with a two-dose regimen as the dependent variable and age, sex, grade, past history of allergic reaction (due to food, medication, animal, pollen from plants and trees, house dust mite, unknown allergen), anaphylaxis, asthma, atopic dermatitis, and attitudes toward COVID-19 vaccination (Q1–Q14) as independent variables. Multivariate logistic regression analysis using a forward selection method was carried out with willingness to receive COVID-19 vaccination in the future (Q15) as the dependent variable and the same factors as independent variables. **Abbreviations**: COVID-19, Coronavirus disease 2019; CI, Confidence interval.

**Table 4 vaccines-09-01295-t004:** Factors associated with changes in willingness to receive the COVID-19 vaccine in the future.

	Odds Ratio	95%		CI	Wald Value	*p* Value
Group A students who were unvaccinated or dropped out after the first dose of the COVID-19 vaccine but who are willing to receive the vaccine in the future.						
Q10	4.50	1.69	-	11.96	9.11	0.003
Q14	0.31	0.13	-	0.74	6.95	0.008
Group B students who received a second dose of the COVID-19 vaccine but are not willing to receive the vaccine in the future.						
Grade	0.58	0.43	-	0.79	11.77	0.001
Q2	3.23	1.44	-	7.25	8.12	0.004
Q4	2.95	1.23	-	7.09	5.83	0.016
Q12	0.26	0.08	-	0.93	4.33	0.038

Multivariate logistic regression analysis using a forward selection method was carried out with changed willingness to receive COVID-19 vaccination in future (Groups A and B) as the dependent variable and age, sex, grade, past history of allergic reaction (due to food, medication, animal, pollen from plants and trees, house dust mites, unknown allergen), anaphylaxis, asthma, atopic dermatitis, and attitudes toward COVID-19 vaccination (Q1–Q14) as independent variables. **Abbreviations**: COVID-19, Coronavirus disease 2019; CI, Confidence interval.

**Table 5 vaccines-09-01295-t005:** Factors associated with willingness to receive a third dose of the COVID-19 vaccine among medical students.

	Odds Ratio	95%		CI	Wald Value	*p* Value
Grade	1.41	1.16	-	1.70	11.93	0.001
Q1	9.85	1.08	-	89.83	4.11	0.043
Q9	2.18	1.19	-	4.01	6.29	0.012
Q10	0.32	0.17	-	0.57	14.61	<0.001
Q12	5.83	2.53	-	13.43	17.18	<0.001
Q14	2.20	1.20	-	4.02	6.53	0.011

Multivariate logistic regression analysis using a forward selection method was carried out with willingness to receive a third dose of the COVID-19 vaccine in the future (Group C) as the dependent variable and age, sex, grade, past history of allergic reaction (due to food, medication, animal, pollen from plants and trees, house dust mites, unknown allergen), anaphylaxis, asthma, and atopic dermatitis, and attitudes toward COVID-19 vaccination (Q1-Q14) as independent variables. **Abbreviations**: COVID-19, Coronavirus disease 2019; CI, Confidence interval.

## Data Availability

The Ethics Committee of Dokkyo Medical University School of Medicine has set restrictions on data sharing because the data contain potentially identifying or sensitive student information. Please contact the institutional review board of the Ethics Committee of Dokkyo Medical University School of Medicine for data requests. Upon request, the ethics committee will decide whether to share the data.

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
