# Peer review of "Attitudes of Medical Students toward COVID-19 Vaccination: Who Is Willing to Receive a Third Dose of the Vaccine?"

_vaccines, 2021, doi:10.3390/vaccines9111295_

Round 1
Reviewer 1 Report
In this study, Sugawara et al. assessed the factors affecting attitudes of professional and young individuals, medical students, toward COVID-19 vaccination especially a third dose. The paper was nicely designed, the presentation was clear and concise and the manuscript was well written. A few points require better revision as commented.
1.Page 2, paragraph 3 in the introduction, the authors presented that the rationale of a third dose is to reduce the risk of infection and reverse neutralizing antibody fading. It is better to introduce the best timing of a third dose after two doses and whether a third dose is recommended for all approved vaccines from different manufacturers.
2.In Table 1, it is surprising that 74.2% of medical students selected "strongly agree" or "agree" for Q8 and believed that the COVID-19 vaccine showed serious side effects. This number is far higher than that from e.g. Comirnaty clinical trials: very common adverse reactions >1/10; common >1/100 to <1/10; uncommon >1/1000 to <1/100. Were the side effects really serious? Consider all medical students having sufficient medical knowledge.
3.It is likely better to show the data of Figure 1 in pie chart showing percentages of every group.
4.Page 7, first paragraph, Table 5 was missing in the text. It is better to present that willingness is associated with vaccine efficacy (Q9, Q14) and benefit (Q12).
5.Page 7, second paragraph in the discussion, the authors mentioned that the rates of vaccine hesitancy in the general population and medical students were comparable, which was difficult to be explained. It is assumed that the authors will find the difference when they compare medical students with the population aged 20 to 24 instead of general population.
6.Page 8, second paragraph, the reason that side effects were not associated with willingness may be that the side effects were not severe. Combining this with point number 2 above, I do not think that the approved vaccine had a safety issue.
7.Page 8, third paragraph, the associated found in Q4 may be related to a cultural issue.
Author Response
> 1. Page 2, paragraph 3 in the introduction, the authors presented that the rationale of a third dose is to reduce the risk of infection and reverse neutralizing antibody fading. It is better to introduce the best timing of a third dose after two doses and whether a third dose is recommended for all approved vaccines from different manufacturers.
According to the suggestion, we searched the literature concerning the best timing of a third dose or boosting with heterologous vaccines. We concluded that these points are controversial in the current situation. Therefore, we added a description and references concerning this situation.
> 2.In Table 1, it is surprising that 74.2% of medical students selected "strongly agree" or "agree" for Q8 and believed that the COVID-19 vaccine showed serious side effects. This number is far higher than that from e.g. Comirnaty clinical trials: very common adverse reactions,,,
In the original Japanese questionnaire, the expression has an attenuated meaning. It is a problem in translation. Therefore, we have changed the expression in our revised manuscript.
> 3. It is likely better to show the data of Figure 1 in pie chart showing percentages of every group.
In the manuscript, we intend to show the association between the real status of COVID-19 vaccination and willingness to receive the vaccine in the future. Therefore, we believe that the current figure is better than the pie chart.
> 4. Page 7, first paragraph, Table 5 was missing in the text. It is better to present that willingness is associated with vaccine efficacy (Q9, Q14) and benefit (Q12).
Thank you for your help. We added the description of table 5. However, a description concerning each item is previously mentioned in the results section. To avoid repetitive description, we would not obey the latter half of the suggestion.
> 5.Page 7, second paragraph in the discussion, the authors mentioned that the rates of vaccine hesitancy in the general population and medical students were comparable, which was difficult to be explained. It is assumed that the authors will find the difference when they compare medical students with the population aged 20 to 24 instead of general population.
According to the reviewer’s viewpoint, we rewrote the description.
> 6. Page 8, second paragraph, the reason that side effects were not associated with willingness may be that the side effects were not severe. Combining this with point number 2 above, I do not think that the approved vaccine had a safety issue.
As mentioned previously, it is a problem in translation. Therefore, we have changed the expression in our revised manuscript.
> 7. Page 8, third paragraph, the associated found in Q4 may be related to a cultural issue.
Thank you for your help. We added the description of cultural issue.
Reviewer 2 Report
REPORT ON VACCINES-1433913-peer review
The introduction of this manuscript affords an illustration of the social, medical, and psychological effects of COVID-19. The isolation effects and the teleworking expansion are producing psychological diseases and are having deep effect on the education systems, those on medical students being the most important for their consequence on the fight itself against
the infection spreading throughout the world.
The statistical analysis of the opinion of medical students about their consent to a third vaccination led the authors of this manuscript to very interesting results. The large majority of
medical students are in favor of vaccination. This is a positive sign, because the opinion of medical students in favor of vaccination may convince many more people to accept vaccination, an important condition for a successful fight against COVID-19.
I think that this is an interesting manuscript deserving publication as soon as possible.
I have a suggestion for a possible improvement of the manuscript. This has to do with a result of this manuscript. In the conclusion of this manuscript, the authors point out that the hesitation to a third vaccination may reflect concern about mobility restrictions. Is this a positive aspect as far the fight against COVID-19 is concerned? The mobility restrictions may have bad effects on the economy, and it would be interesting to establish why the concern about mobility restrictions generates hesitancy to adopt a third vaccination rather that than increasing the willingness to do it. If the authors have a plausible explanation, they may add a few lines to the conclusion. If they do not, they may ignore my question.
Author Response
> In the conclusion of this manuscript, the authors point out that the hesitation to a third vaccination may reflect concern about mobility restrictions. Is this a positive aspect as far the fight against COVID-19 is concerned? The mobility restrictions may have bad effects on the economy, and it would be interesting to establish why the concern about mobility restrictions generates hesitancy to adopt a third vaccination rather that than increasing the willingness to do it. If the authors have a plausible explanation, they may add a few lines to the conclusion. If they do not, they may ignore my question.
In our manuscript, the association we found is between relaxation of mobility restrictions and willingness to receive the third dose of the COVID-19 vaccine. We do not state that hesitation for a third vaccination may reflect concern about mobility restrictions in conclusion section. Therefore, we would not obey the suggestion.
Reviewer 3 Report
The authors reported the attitudes of medical students to the Covid-19 vaccination. Overall, this work provides some meaningful information about the Covid-19 vaccination.
Major comments:
- The study participants were from one medical university, and just 66.8% (496/742) of the students in this university completed the questionnaire. The responding rate is not high. It is unknown whether the participants may basically represent the overall situation. It should be a limitation.
- The detailed information of participants is not adequately presented. A full medical course in Japan is 6 years. Because the students in grade 5 and 6 are quite different from those in grade 1 and 2 in terms of medical knowledge and contacting patients, it is better to present the numbers of students and those who responded the questionnaires in each grade.
- It is not adequate to have grouped the students who received one vaccine dose and who did not receive the vaccine at all in the same group (Fig. 1), because they have different attitude to the Covid-19 vaccination. The reasons for drop out after the first dose should be listed, and it is better to present the reasons for the decline of Covid-19 vaccination.
Other comments:
The statement that “more than 80% (648/742) of medical students completed a two-dose regimen of the Comirnaty vaccine between April and May in 2021” requires a reference or the data source.
Table 1, all the denominators are 496, and they can be omitted and by adding 496 in the table title.
In the Results section, the first sentence “Among all the participants, 90.7% indicated that they …”. It is better to add the frequency after the percentage “90.7% (450/496)”, because the Table 1 does not directly contain such data.
Table 1, it appears to be difficult to explain that 74.2% (368/496) of the students considered that the Covid-19 vaccine has serious side effects, but most of them still received the vaccination.
Table 2 is difficult to understand. Were some data mistakenly entered?
Table 3, the footnote, the detailed questions of Q1, Q4, … are repeated, and do not require to present here, but by modifying: “the detailed questions of Q1, Q4, … are presented in Table 1”. Similarly in Table 4 and Table 5.
Discussion: “Fear of vaccination side effects has been reported to be associated with COVID-19 vaccine hesitancy in medical students as well as the general population [9, 26, 29]”. A recently published article in Vaccines also revealed that worry about the vaccine safety is associated with vaccine hesitancy [Xu B, Gao X, Zhang X, Hu Y, Yang H, Zhou YH. Real-World Acceptance of COVID-19 Vaccines among Healthcare Workers in Perinatal Medicine in China. Vaccines (Basel). 2021;9(7):704. doi: 10.3390/vaccines9070704.].
Author Response
Major comments:
> 1. The study participants were from one medical university, and just 66.8% (496/742) of the students in this university completed the questionnaire. The responding rate is not high. It is unknown whether the participants may basically represent the overall situation. It should be a limitation.
We added the description of limitation.
> 2. The detailed information of participants is not adequately presented. A full medical course in Japan is 6 years. Because the students in grade 5 and 6 are quite different from those in grade 1 and 2 in terms of medical knowledge and contacting patients, it is better to present the numbers of students and those who responded the questionnaires in each grade.
We added the response rate of each grade to the methods section.
> 3. It is not adequate to have grouped the students who received one vaccine dose and who did not receive the vaccine at all in the same group (Fig. 1), because they have different attitude to the Covid-19 vaccination. The reasons for drop out after the first dose should be listed, and it is better to present the reasons for the decline of Covid-19 vaccination.
In our study population, two students received only one vaccine dose. Unfortunately, the reasons for dropout after the first dose are not clear. However, one student among the two was willing to receive the vaccine in the future (Q15). Grouping the students who received one vaccine dose and who did not receive the vaccine at all in the same group might not be adequate. However, after the deletion of data regarding the dropout students, we obtained comparable results of tables. Therefore, we want to publish the tables and figures in the current form.
Other comments:
> The statement that “more than 80% (648/742) of medical students completed a two-dose regimen of the Comirnaty vaccine between April and May in 2021” requires a reference or the data source.
It is based on the Health Services Center for Students and Staff (HSCSS) database.
> Table 1, all the denominators are 496, and they can be omitted and by adding 496 in the table title.
According to the reviewer’s suggestion, we changed the description in Table 1.
> In the Results section, the first sentence “Among all the participants, 90.7% indicated that they …”. It is better to add the frequency after the percentage “90.7% (450/496)”, because the Table 1 does not directly contain such data.
Thank you for your help. We added the frequency in the results section.
> Table 1, it appears to be difficult to explain that 74.2% (368/496) of the students considered that the Covid-19 vaccine has serious side effects, but most of them still received the vaccination.
In the original Japanese questionnaire, the expression has an attenuated meaning. It is a problem in translation. Therefore, we have changed the expression in our revised manuscript.
> Table 2 is difficult to understand. Were some data mistakenly entered?
Thank you very much for your help. We corrected the mistake.
> Table 3, the footnote, the detailed questions of Q1, Q4, … are repeated, and do not require to present here, but by modifying: “the detailed questions of Q1, Q4, … are presented in Table 1”. Similarly in Table 4 and Table 5.
According to the reviewer’s suggestion, we deleted the description.
> Discussion: “Fear of vaccination side effects has been reported to be associated with COVID-19 vaccine hesitancy in medical students as well as the general population [9, 26, 29]”. A recently published article in Vaccines also revealed that worry about the vaccine safety is associated with vaccine hesitancy [Xu B, Gao X, Zhang X, Hu Y, Yang H, Zhou YH. Real-World Acceptance of COVID-19 Vaccines among Healthcare Workers in Perinatal Medicine in China. Vaccines (Basel). 2021;9(7):704. doi: 10.3390/vaccines9070704.].
We referred the article in our manuscript.
Reviewer 4 Report
This research article by Sugawara. et al. investigated medical students’ attitudes toward COVID-19 vaccination, especially for a third dose of the vaccine through assessment of factors associated with their attitudes. The paper demonstrated that willingness to receive a third dose of vaccine associated with some factors including grade, positive attitude toward vaccination, belief in the protection offered by vaccination, concern about the sustainability of immunity. Purpose of the research is clear, the manuscript is well written, questions in survey are fine, and the data were analyzed appropriately. However, some publications have covered the similar population (medical students). In addition, the scale of the study is relatively small and the participants were medical students in one university, which limited impacts of the study. To conclude, the article provides meaningful information, and it fits well to the scope of this journal, but I believe that additional data will benefit the readers of this journal. I will not recommend the publication of the manuscript with this current form. The comments to consider in subsequent versions are shown below.
- Authors could consider increasing the scale of the study through recruitment of medical students in some universities.
- Page 7 Line 23 “Regarding medical students, the rate of vaccine hesitancy reported in previous studies has varied considerably due to the different definitions of the concept or timing of investigation in each study.” I agree that timing and environments largely affect rate of vaccination hesitancy. Authors could consider categorizing the environments in this and the published papers according to availability of vaccination, the percentage of vaccinated students or maybe other factors to identify environmental factors associated with vaccination hesitancy. Authors also could consider expanding targets to the students who have majors in health-care associated programs and will contrast factors associated with vaccination hesitancy among the population in Japan.
Author Response
> Authors could consider increasing the scale of the study through recruitment of medical students in some universities.
Although increasing the scale of our study might be attractive, such choices are time-consuming. Whether to receive a third dose of the COVID-19 vaccine is currently an issue. We believe that our work deserves publication in the current dataset.
> Page 7 Line 23 “Regarding medical students, ,,,.” I agree that timing and environments largely affect rate of vaccination hesitancy. Authors could consider categorizing the environments in this and the published papers according to availability of vaccination, the percentage of vaccinated students or maybe other factors to identify environmental factors associated with vaccination hesitancy.
We added a description concerning environmental factors for vaccination hesitancy.
Round 2
Reviewer 3 Report
The authors have addressed all questions. The revised manuscript has been sufficiently improved and it is now acceptable for publication in Vaccines.
Reviewer 4 Report
Dear Authors,
Your revision responded to the questions raised by reviewers and the manuscript was improved. I recommend to accept the paper in present form.